**EMBO**
*reports*

# Exploring GPCR-arrestin interfaces with genetically encoded crosslinkers

Thore Böttke[1] iD, Stefan Ernicke[1], Robert Serfling[1], Christian Ihling[2], Edyta Burda[3],
Vsevolod V Gurevich[4] iD, Andrea Sinz[2] & Irene Coin[1],* iD

## Abstract

β-arrestins (βarr1 and βarr2) are ubiquitous regulators of G protein-coupled receptor (GPCR) signaling. Available data suggest that β-arrestins dock to different receptors in different ways. However, the structural characterization of GPCR-arrestin complexes is challenging and alternative approaches to study GPCR-arrestin complexes are needed. Here, starting from the finger loop as a major site for the interaction of arrestins with GPCRs, we genetically incorporate non-canonical amino acids for photo- and chemical crosslinking into βarr1 and βarr2 and explore binding topologies to GPCRs forming either stable or transient complexes with arrestins: the vasopressin receptor 2 (rhodopsin-like), the corticotropin-releasing factor receptor 1, and the parathyroid hormone receptor 1 (both secretin-like). We show that each receptor leaves a unique footprint on arrestins, whereas the two β-arrestins yield quite similar crosslinking patterns. Furthermore, we show that the method allows defining the orientation of arrestin with respect to the GPCR. Finally, we provide direct evidence for the formation of arrestin oligomers in the cell.

**Keywords** G protein-coupled receptor; genetically encoded crosslinkers; GPCR–arrestin complexes; live cells; β-arrestins

**Subject Categories** Methods & Resources; Signal Transduction; Structural Biology

## Introduction

Arrestins are a small family of highly homologous cytosolic proteins that dock to activated and phosphorylated G protein-coupled receptors (GPCRs) to desensitize G protein-mediated signaling (Gurevich, 2014; Shukla & Dwivedi-Agnihotri, 2020). Four arrestin isoforms are expressed in vertebrates: Two quench the signaling by rhodopsin (Rho) and cone opsins in the retina (arr1 and arr4, also called

"visual" arrestins), whereas the other two serve ubiquitously the hundreds of non-visual GPCRs encoded in the human genome (arr2 and arr3, a.k.a. βarr1 and βarr2, called "non-visual" or β-arrestins). All arrestins in their basal state consist of β-sheets organized in two cup-like domains (N- and C-domain) with four exposed loops in the central crest of the receptor binding side (Fig 1A). The *finger loop* shares a consensus motif with the C-terminal helix of Gα from Gi/transducin and competes with it for the cavity that opens in the transmembrane (TM) core of active GPCRs (Szczepek *et al*, 2014). This loop is indispensable for the formation of a fully engaged arrestin–receptor complex (core conformation). Arrestin variants lacking the *finger loop* only bind to the phosphorylated GPCR C-terminal tail via their N-domain (tail conformation) (Shukla *et al*, 2014; Cahill *et al*, 2017; Nguyen *et al*, 2019). Moreover, the *finger loop* determines the binding preferences of arrestin to receptors (Vishnivetskiy *et al*, 2004, 2011; Chen *et al*, 2017).

G protein-coupled receptors are divided into two classes with respect to arrestin binding (Oakley *et al*, 1999, 2000; Luttrell & Lefkowitz, 2002). Class A receptors form transient and rapidly dissociating complexes with arrestin, and resensitize rapidly. These receptors interact with both β-arrestins, but show a bias toward βarr2. Besides the prototypical β2-adrenergic, class A receptors include, among others, the μ opioid, endothelin A, and dopamine D1A receptors (all rhodopsin-like GPCRs), as well as the corticotropin-releasing factor receptor (CRF1R, secretin-like) (Oakley *et al*, 2007; Grammatopoulos, 2012). Class B receptors form long-lived complexes with arrestin that remain stable through the internalization via clathrin-coated pits, and resensitize slowly. Class B receptors bind with high affinity either β-arrestin. The prototypic class B arrestin binder is the vasopressin 2 receptor (V2R). Other receptors forming stable complexes with arrestin include the angiotensin II type 1 receptor (AT1R), the oxytocin receptor, the neurotensin 1 receptor (NTS1R), and the secretin-like parathyroid hormone receptor (PTH1R) (Oakley *et al*, 2001; Vilardaga *et al*, 2002). The latter has been shown to form GPCR–arrestin–G protein megaplexes that mediate prolonged signaling after internalization in endosomes (Wehbi *et al*, 2013; Thomsen *et al*, 2016). A major determinant for the stability of GPCR-arrestin complexes is the

1   Institute of Biochemistry, Faculty of Life Sciences, University of Leipzig, Leipzig, Germany
2   Institute of Pharmacy, Department of Pharmaceutical Chemistry and Bioanalytics, Charles Tanford Protein Center, Martin Luther University Halle-Wittenberg, Halle/Saale, Germany
3   Institute of Pharmacy, Faculty of Medicine, University of Leipzig, Leipzig, Germany
4   Department of Pharmacology, Vanderbilt University, Nashville, TN, USA
    *Corresponding author. Tel: +49 (0)341 9736996; Fax: +49 (0)341 9736909; E-mail: irene.coin@uni-leipzig.de

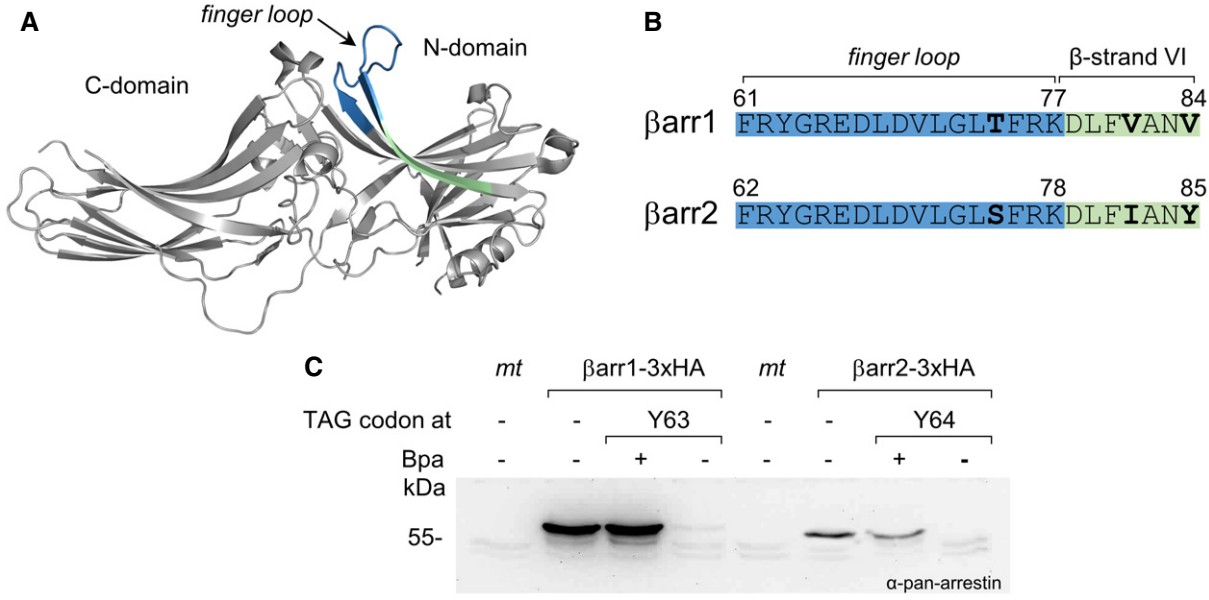

**Figure 1. Genetic incorporation of Bpa into β-arrestins.**

A   Ribbon representation of bovine βarr1 (PDB 1G4M) (Han *et al*, 2001). The *finger loop* is highlighted in blue and β-strand VI in pale green.
B   Sequence alignment of the *finger loop* and N-terminal part of β-strand VI in bovine β-arrestins, color coded as in (A). Divergent positions are shown in bold.
C   Western blots of the lysates of HEK293T cells, resolved on SDS–PAGE and stained with a pan-arrestin antibody. *mt*, mock transfected. Bpa: benzoyl-phenylalanine. Arrestins were equipped with a 3xHA tag at the C-terminus, which increases their size of about 3 kDa compared to the endogenous proteins (βarr1 47 kDa, βarr2 46 kDa). βarr1 and βarr2 were transfected at 1/3 DNA compared to the corresponding TAG-mutants.

C-terminal tail of the receptor. In general, GPCRs carrying clusters of Set/Thr in the C-terminal tail show a class B behavior (Oakley *et al*, 1999).

Overall, GPCR-arrestin complexes are highly dynamic and the biophysical determination of their structure is technically very challenging. To date, only a few 3D structures of arrestin–GPCR complexes have been solved: the crystal structure of the rhodopsin (Rho)-bound arr1 (Kang *et al*, 2015; Zhou *et al*, 2017), and the three very recent cryo-electron microscopy (cryo-EM) structures of βarr1 in complex with the class B NTS$_1$R (Yin *et al*, 2019; Huang *et al*, 2020), the muscarinic acetylcholine-2-receptor (M$_2$R) (Staus *et al*, 2020), and the β$_1$-adrenoceptor (β$_1$-AR) (Lee *et al*, 2020). The β$_1$-AR belongs to class A arrestin binders (Shiina *et al*, 2000, Eichel *et al*, 2016), whereas contradictory findings have been reported for the M$_2$R (Gurevich *et al*, 1995, Jones *et al*, 2006).

As expected on the basis of numerous biochemical studies (reviewed in (Gurevich & Gurevich, 2006)), the existing structures confirm that βarr1 can assume distinct orientations depending on which receptor it binds to. Likewise, position and conformation of the *finger loop* differ between the complexes. Moreover, the binding mode of arrestin to the same GPCR can be affected by specific phosphorylation patterns, which lead to different downstream responses (barcode hypothesis) (Tobin *et al*, 2008; Nobles *et al*, 2011; Yang *et al*, 2015; Zhou *et al*, 2017; Mayer *et al*, 2019; Kaya *et al*, 2020). These facts, together with the observation that there are no conserved motifs in the intracellular part of the GPCRs, suggest that homology models based on the few available structures likely have limited predictive value. In addition, the lipid environment has been shown to affect the conformation of isolated GPCR-arrestin

complexes (Staus *et al*, 2020). It is also worth mentioning that the structures of M$_2$R and β$_1$-AR could only be achieved by equipping the receptors with the C-terminal tail of the V$_2$R, a modification that stabilizes the GPCR-arrestin complexes by transforming class A arrestin binders into class B ones (Oakley *et al*, 2000). Methods are needed both to address complexes that elude direct structural characterization and to validate biophysical data in the living cell.

We have a long experience in the application of genetically encoded chemical tools to map interactions of GPCRs with their ligands (Coin *et al*, 2011, 2013; Seidel *et al*, 2017), as well as interactions of nuclear receptors (Schwarz *et al*, 2013, 2016). Here we show that the incorporation of crosslinking non-canonical amino acids (ncAAs) into arrestins allows the comparison of the topology of GPCR-arrestin complexes in intact cells.

## Results and Discussion

We selected 24 positions in βarr1 and βarr2 covering the finger loop italic and the N-terminal segment of the following β-strand (Fig 1B). The codons of each amino acid along this stretch were substituted with the amber stop codon TAG for the incorporation of the photoactivatable ncAA benzoyl-Phe (Bpa). A preliminary test on a pair of homologous βarr1/βarr2 mutants showed that Bpa was smoothly incorporated into both arrestins via amber suppression (Wang, 2017a; Lang *et al*, 2018), with minimal read-through occurring in the absence of the ncAA (Fig 1C). Transfected arrestins were expressed in very large excess compared to the endogenous counterparts. βarr1 is known to express at a higher level than βarr2 in

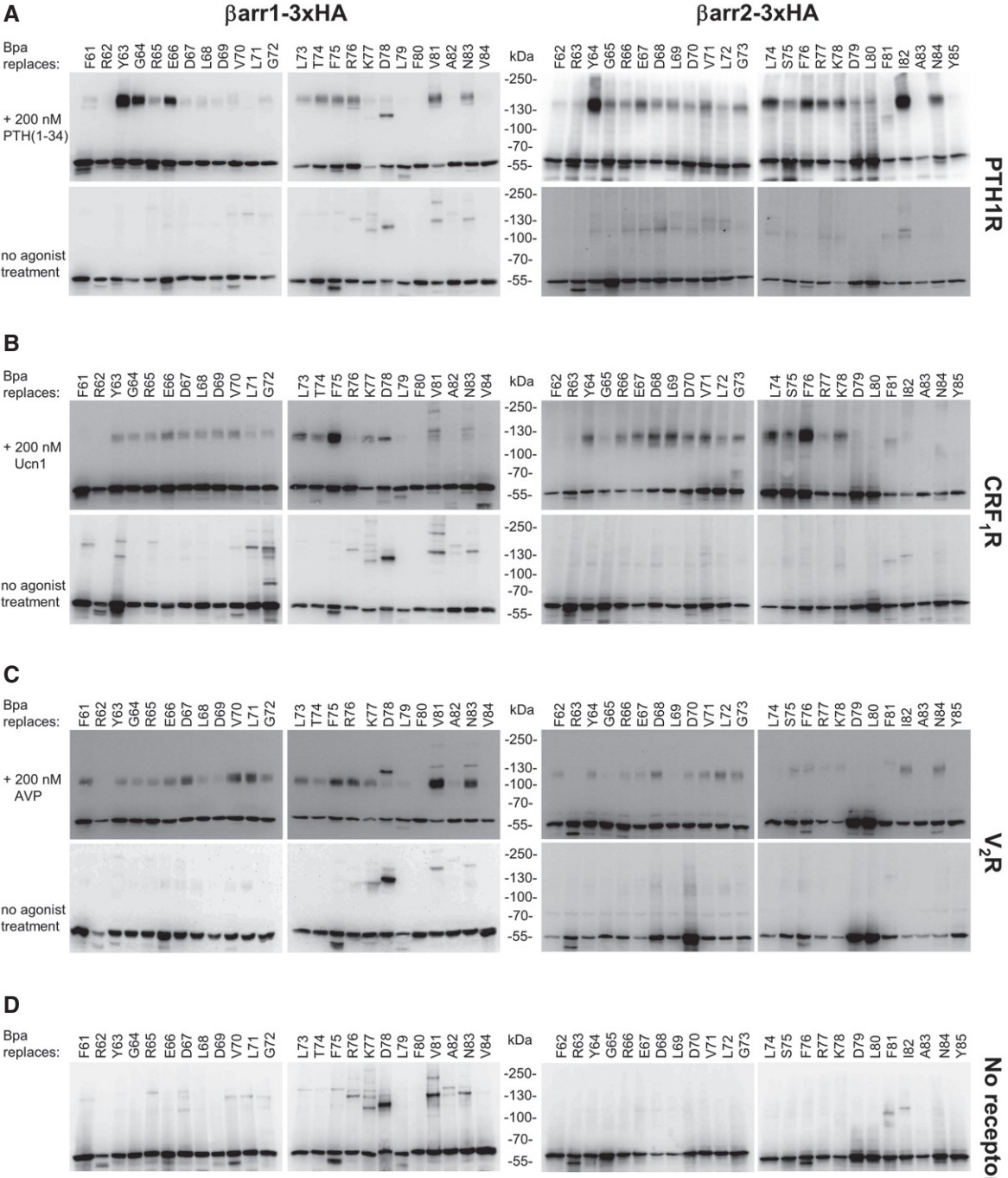

**Figure 2. Photo-crosslinking of Bpa-arrestins with three different GPCRs with and without agonist stimulation.**

Each panel represents immunoblots of whole cell lysates stained with an anti-HA antibody detecting arrestin. In the left panels, GPCRs were combined with βarr1 variants and in the right panels with βarr2 variants. Residues replaced by Bpa are indicated above each lane. Covalent arrestin–receptor complexes are expected between 100–200 kDa, considering the size of the receptors ($V_2R$ 41.5 kDa, PTH1R 63.8 kDa, $CRF_1R$, 46.3 kDa) and their glycosylation.

A   PTH1R, agonist PTH(1–34).
B   $CRF_1R$, agonist Urocortin 1 (Ucn1).
C   $V_2R$, agonist arginine-vasopressin (AVP).
D   No receptor.

native cells and different cell lines (Gurevich *et al*, 2004; Coffa *et al*, 2011). This can be observed also here, both in the case of endogenous arrestins and the Bpa-mutants, which expressed at a homogenous yield (Fig 2). The Bpa-mutation did not hamper the recruitment of either arrestin to the PTH1R receptor, suggesting that the overall functionality of the arrestins is preserved (Appendix Fig S1).

The two sets of Bpa-βarr1 and Bpa-βarr2 were combined with GPCRs forming either stable (class B) or transient (class A) complexes with arrestins. We selected two class B receptors belonging to two distinct phylogenetic GPCR families, the V$_2$R (rhodopsin-like) and the PTH1R (secretin-like), as well as the class A receptor CRF$_1$R (secretin-like). The latter was preferred to other class A GPCRs because the CRF system is well established in our laboratory. Photo-crosslinking experiments were carried out in live HEK293T cells expressing one receptor and one arrestin variant, for a total of 144 combinations, with or without agonist stimulation (Fig EV1). Cell lysates were resolved on SDS–PAGE and immunoblotted with antibodies detecting either arrestin (Fig 2) or the GPCR (Fig EV2). For a subset of arrestin–receptor combinations, high molecular weight (MW) bands were observed (Fig 2A–C upper panels). Most bands appeared only when arrestin recruitment was triggered by the agonist treatment, but were not visible when the receptor was not activated (Fig 2A–C, lower panels) or absent (Fig 2D). The apparent MW was receptor-dependent and was reduced by deglycosylation (Fig 3). These activation-dependent bands were therefore attributed to covalently crosslinked receptor–arrestin complexes. The crosslinking patterns differed substantially between the different receptors, whereas the strongest bands with the same receptor appeared at homologous Bpa positions in the two arrestins. The most prominent bands were obtained with Bpa63/64-βarr1/2 (crosslinker at the beginning of the *finger loop*) with the PTH1R, Bpa75/76-βarr1/2 (crosslinker at the C-terminal portion of the *finger loop*) with the CRF$_1$R and Bpa81/82-βarr1/2 (crosslinker in β-strand VI) with the V$_2$R. Overall, these results suggest specific binding modes of arrestins at different receptors, whereas the two arrestins appear to bind to each receptor in a similar fashion.

These findings confirm the observation from the 3D structures, which show distinct binding modes of βarr1 to the different receptors. In a recent report, photo-crosslinkers genetically incorporated into the intracellular domains of the AT$_1$R have revealed distinct binding modalities of AT$_1$R to βarr1 depending on the type of the agonist used for its activation (natural angiotensin vs. arrestin-biased AT$_1$R ligands) (Gagnon *et al*, 2019). Collectively, these results demonstrate that genetically encoded photo-crosslinkers incorporated either into a GPCR or into arrestin allow elucidating with a good sensitivity differences in the arrangement of arrestin–GPCR complexes.

Other crosslinking bands observed in our experiments appeared independently from activation of a GPCR. The most prominent activation-independent bands were observed with Bpa78- and Bpa81-βarr1 at apparent MW ~ 120 kDa. They appeared in all analyses (Fig 2A–C) even in the absence of a co-transfected receptor (Fig 2D) and were not affected by deglycosylation (Fig 3). Only very faint bands were observed with the Bpa-βarr2 set.

To identify the nature of these crosslinking products, Bpa78-βarr1 was equipped with a 2xStrep tag at the C-terminus, expressed in large scale in HEK293T cells, and isolated via the Strep-Tactin® XT system either before or after UV irradiation (Appendix Fig S2A). As shown in Fig 4, the crosslinking product obtained by *in vitro* irradiation of the isolated Bap78-βarr1 runs exactly as the crosslinking product obtained when irradiating the intact cells (Fig 4A). Furthermore, the crosslinked samples were subjected to size exclusion chromatography (SEC), yielding two partially resolved major peaks (Fig 4B). Eluates were analyzed by immunoblotting (Fig 4C) and by MALDI mass spectrometry (MS, Fig 4D). The peak at higher retention volume (V$_R$3) yielded a single signal in Western blot and a whole-protein mass spectrum compatible with a βarr1 monomer (50.4 kDa). The other eluted fraction (V$_R$2) contained a mixture of βarr1 with the crosslinking product, and featured in MS a further signal at double mass. Furthermore, SEC fractions were enzymatically digested and applied to nano-HPLC/nano-ESI-Orbitrap-MS/MS. In the SEC fraction containing the crosslinked product, βarr1 was identified as the by far most abundant protein with sequence

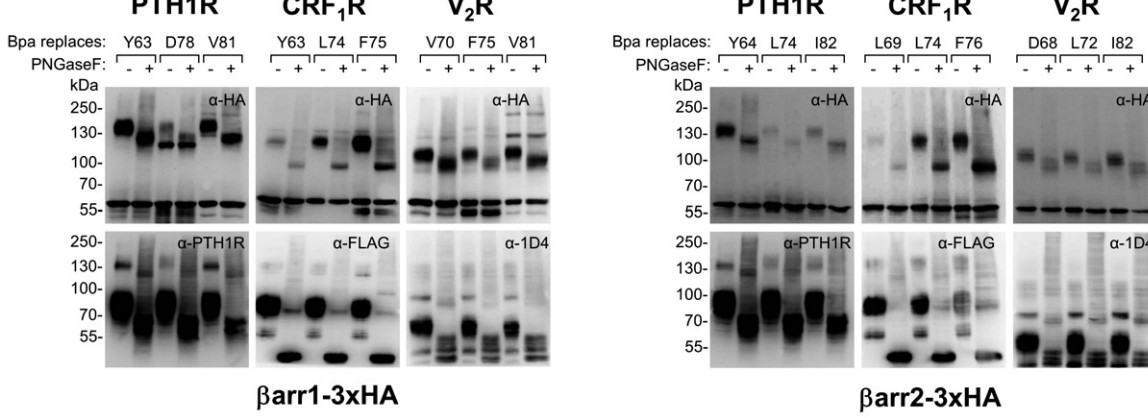

**Figure 3. Response of photo-crosslinked samples to deglycosylation.**
Aliquots from agonist-treated samples shown in Fig 2 were deglycosylated with PNGaseF, run on SDS–PAGE and immunoblotted with antibodies detecting arrestin (upper panels, α-HA) and the GPCRs (as indicated, CRF$_1$R is equipped with a N-terminal FLAG, V$_2$R carries a C-terminal 1D4 epitope).

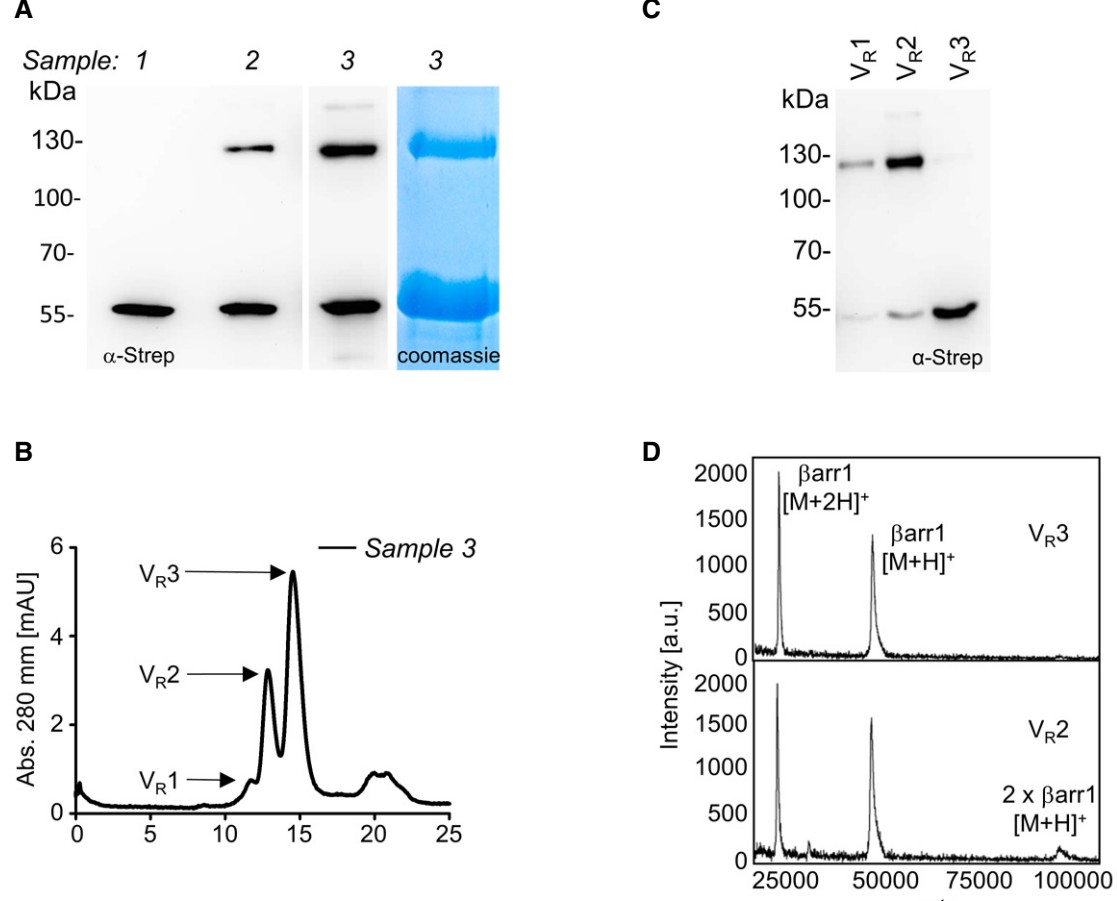

**Figure 4.   Analysis of activation-independent crosslinking band at D78Bpa-βarr1.**

A   SDS–PAGE and immunoblots of immunoprecipitated samples. Sample 1 was immunoprecipitated without previous UV treatment. Sample 2 is the product of irradiation of sample 1. Sample 3 was treated with UV light before immunoprecipitation. All samples were run in parallel on the same gel.
B   Size exclusion chromatography (SEC) of Sample 3.
C   Western blot of the fractions eluted by SEC.
D   MALDI-TOF analysis of the SEC fractions $V_R2$ and $V_R3$. The spectrum of sample $V_R2$ shows a $[M + H]^+$ peak featuring twice the mass (~ 108,000 Da) of the $[M + H]^+$ peak of the βarr1 monomer (sample $V_R3$, ~ 50,400 Da).

coverage of 90% (Appendix Fig S2B) and 1,951 PSMs (peptide spectral matches) compared to only 22 PSMs for the second most abundant protein. Collectively, the results identify this crosslinking product as a covalent βarr1 dimer.

β-arrestins are known to self-associate in solution (Hanson *et al*, 2008; Chen *et al*, 2014), as well as in crystals and in cells (Han *et al*, 2001; Milano *et al*, 2006; Boularan *et al*, 2007; Zhan *et al*, 2011). Our results support the notion that β-strand VI participates in the βarr1 association interface, which is in line with the current oligomerization model of this arrestin. On the other hand, it cannot be excluded that at least some of the weak activation-independent crosslinking signals at high MW belong to complexes of arrestin with endogenous proteins. Both arrestins are known to function as scaffolds for a wide variety of proteins, with the most prominent examples being kinases like ERKs, JNK3, or other MAPKs (Xiao *et al*, 2007; Song *et al*, 2009), as well as proteins involved in GPCR trafficking, such as clathrin and AP2 (Goodman *et al*, 1996; Laporte

*et al*, 1999). Elucidation of the nature of all receptor-independent crosslinking signals awaits further experiments.

Other receptor-independent bands were visible below the apparent MW of arrestin itself (e.g., Bpa75/76-βarr1/2; Fig 2). Position 75/76 is quite intriguing, because Bpa75/76-βarr1/2 give both a low-MW band and an activation-dependent crosslinking band (especially with CRF₁R, Fig 2). To test whether low-MW bands are due to intra-molecular crosslinking, we applied proximity-induced pairwise crosslinking (Fig 5A) (Xiang *et al*, 2013; Wang, 2017b; Coin, 2018; Seidel *et al*, 2019). The mildly electrophilic ncAA O-(2-bromoethyl)-tyrosine (BrEtY) (Xiang *et al*, 2014) was genetically incorporated at position F75 of βarr1. Simultaneously, F244, which lies in its close proximity in the inactive βarr1 (Fig 5B, left), was mutated to Cys. Indeed, BrEtY75-Cys244-βarr1, but not BrEtY75-βarr1, yielded on Western blot a major band at the exact position of the receptor-independent product generated by UV irradiation of Bpa75-βarr1 (Fig 5C). When arrestin is activated, the *finger loop*

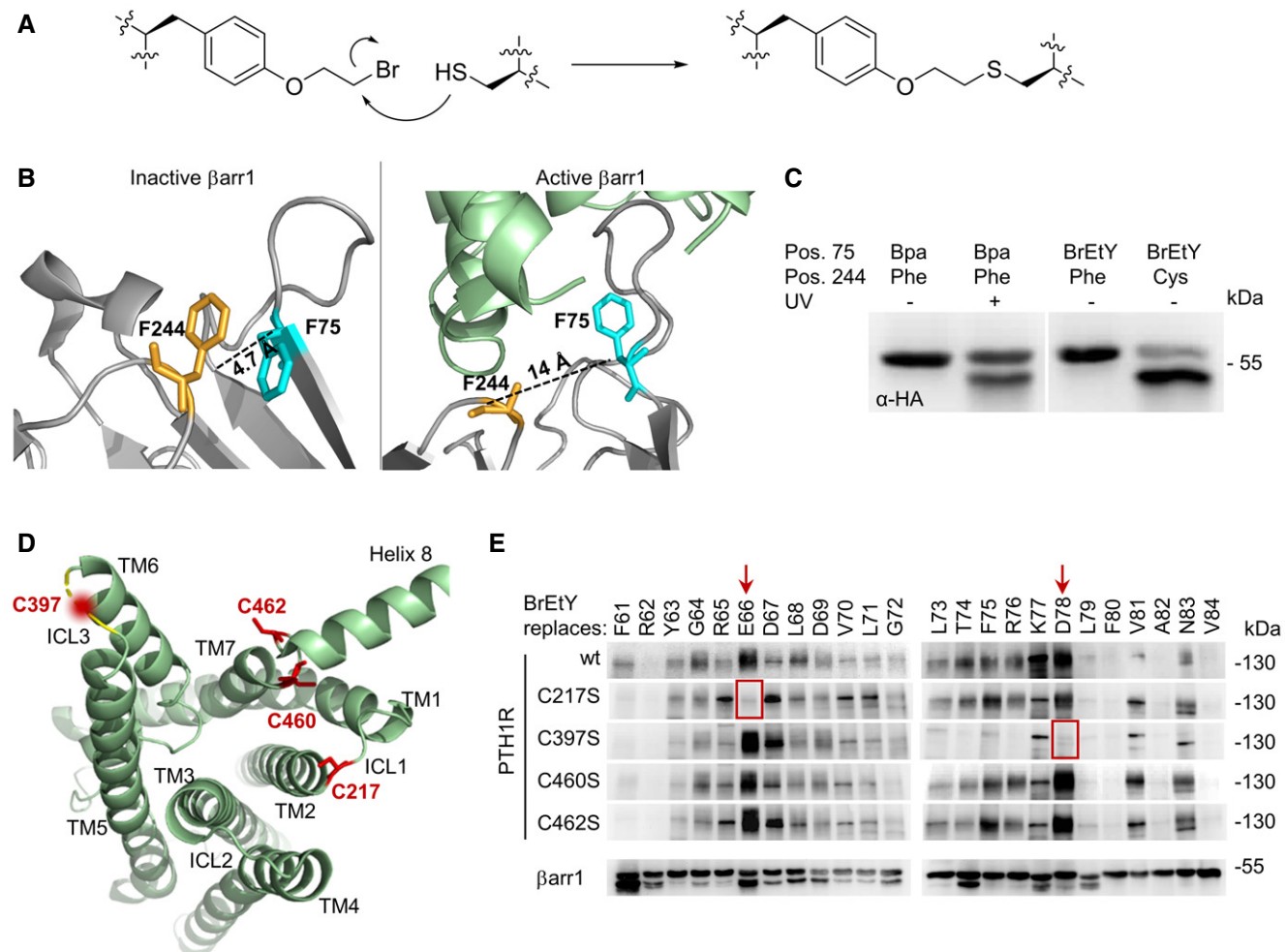

**Figure 5. Proximity-induced pair-wise crosslinking.**

A   Mechanism of the nucleophilic substitution reaction between BrEtY and Cys. The product is a thioether, which is stable in reducing SDS–PAGE.

B   Ribbon representation of inactive (PDB 1G4M (Han *et al*, 2001)) and active (PDB 6U1N (Staus *et al*, 2020)) βarr1. Arrestin is in gray and M₂R in pale green.

C   Western blot of whole cell lysates from HEK293T cells expressing βarr1 mutated as indicated in the top lines. All samples were run in parallel on the same gel.

D, E   Pair-wise crosslinking between BrEtY-βarr1-3xHA and wt PTH1R. (D) Ribbon representation of active PTH1R as seen from the intracellular side (PDB 6NBF) (Zhao *et al*, 2019). Cys residues are shown in red. ECL3 (not resolved in the structure) is represented as a yellow dashed line. The approximate position of the Cys residue in this loop is marked with a fuzzy red circle. (E) Western blot of whole cells lysates stained with an α-HA antibody. Residues of βArr1 exchanged with BrEtY are indicated in the upper row, and mutations at PTH1R are indicated on the left side. The red arrows indicate the two most prominent signals. Red squares indicate signals that vanish upon removal of specific Cys residues in PTH1R.

moves away from F244 to insert into the receptor TM cavity (Fig 5B, right), so that Bpa at position 75 can capture the receptor. Thus, the photo-crosslinking data reveal the co-existence of two distinct populations of arrestins in our experiments: a receptor-engaged population and an inactive population in basal conformation. The result is not surprising, as especially βarr1 is largely overexpressed.

We further explored whether chemical crosslinking can be applied to determine inter-molecular receptor-arrestin proximity points, as it was shown to capture protein–protein interactions in live cells (Yang *et al*, 2017). The PTH1R contains four Cys residues in the juxtamembrane region at the cytosolic side (Fig 5D). To investigate whether any of these residues lie close to the *finger loop* of the associated arrestin, we incorporated BrEtY in the same 24

positions of βarr1 previously substituted with Bpa. BrEtY-βarr1 mutants were co-expressed with PTH1R in HEK293T cells, and arrestin recruitment was triggered by agonist activation. Strong arrestin–receptor crosslinking was detected when BrEtY was incorporated at position E66 and D78, along with some minor signals (Fig 5E). To investigate which Cys reacted with which BrEtY, the four Cys in PTH1R where mutated one by one into Ser and Ser-PTH1R mutants were combined with the same BrEtY-βarr1 set. Crosslinking obtained with BrEtY66- and BrEtY78-βarr1 selectively disappeared upon removal of C217 and C397, respectively (Fig 5E). Overall, the results of chemical crosslinking suggest that in the PTH1R-βarr1 complex, the N-terminal portion of the *finger loop* (E66) points toward ICL1 (C217) of the receptor, whereas the C-

terminal segment (T74-K77, weaker crosslinking signals) up to the beginning of β-strand VI (D78) is proximal to ICL3 (C397). This is compatible with the orientation observed in the arr1-Rho and the βarr1-M$_2$R complex, but not the βarr1-NTS$_1$R (Fig EV3). This result clearly confirms the observation from the existing structures that the orientation of arrestin on the receptor does not determine whether the GPCR–arrestin complexes are stable or transient.

In conclusion, we have devised a biochemical method to investigate topologies of arrestin binding to GPCRs in intact cells, which exploits the genetic incorporation of photo-activatable and mildly electrophilic ncAAs into arrestin for photo- and chemical crosslinking experiments. In the first instance, photo-crosslinking enables distinguishing footprints of different receptors on the same arrestin. We show that different footprints are not related to the phylogenetic class of the investigated GPCRs (rhodopsin-like vs. secretin-like) nor to the type of interaction with arrestin (transient or stable), but are indeed specific for individual GPCRs. The similar footprints obtained on βarr1 and βarr2 suggest a common binding mode for the two arrestins. In the second instance, chemical crosslinking reveals inter-molecular pairs of amino acids coming into proximity in the GPCR-arrestin complex. By exploiting the presence of natural Cys residues in PTH1R, we have defined the orientation of the *finger loop* in the intracellular crevice of the active receptor. We anticipate that the incorporation of BrEtY into arrestins combined with Cys-mutagenesis of the GPCR will allow the detailed investigation of whole GPCR–arrestin interfaces. Overall, once a large set of TAG-arrestin mutants is generated, the method ideally lends itself to the comparison of arrestin binding to several receptors in a short time. We have shown that the approach is applicable both to stable and transient arrestin–receptor complexes, which provides a unique possibility for investigating interactions of arrestin with class A GPCRs without the need of generating chimeras with altered binding behavior. Last, but not least, ncAA-crosslinking provides information directly from the cellular environment, which cannot be obtained by crystallography or cryo-EM.

# Materials and Methods

### Molecular biology

Cloning was performed in *Escherichia coli* DH5α. All PCRs were run with Phusion High Fidelity Polymerase. The ORFs of all arrestins and receptors were cloned into pcDNA3.1 (Thermo Fisher Scientific) using standard cloning methods. Sequences for N- or C-terminal affinity tags were built from scratch using overlapping primers. The primers for the systematic TAG mutagenesis of βarr1 and βarr2 were designed with AA scan (Sun *et al*, 2013), and all changes were made using site directed mutagenesis. Oligonucleotides were purchased by Microsynth (Balgach, CH) and Biomers (Ulm, DE). All sequences were verified by Sanger sequencing (Seqlab Göttingen, DE).

### Plasmids for ncAA incorporation (Appendix Fig S3)

#### pIRE4-Bpa (available from ADDGENE #155342)

Plasmid pIRE4-BpaRS is a bicistronic construct built exactly as pIRE4-Azi (Seidel *et al*, 2017; Serfling *et al*, 2018a). The backbone is

originally based on pEGFP-N1 (Clontech, Mountain View, CA) and bears a Kan/Neo resistance. The CMV-EGFP sequence of pEGFP-N1 was substituted with the AARS cassette, followed by the tRNA cassettes immediately downstream of the polyadenylation sequence. pIRE4-Bpa contains a humanized gene of the *E. coli* BpaRS (Chin *et al*, 2003) (custom synthesized by GeneArt, Thermo Fisher Scientific) under control of a PGK promoter and four tandem repeats of a cassette for the expression of the tRNA suppressor from the *Bacillus stearothermophilus* tRNA$^{Tyr}$ (*Bst*Yam), including a U6 promoter and a 5′-trailer.

#### pNEU-MmXYRS-4xM15 (available from ADDGENE #155343)

pNEU-MmXYRS-4xM15 is a bicistronic vector. The pNEU backbone is essentially the same as pcDNA3.1 with some variations in the restriction sites. The plasmid contains a humanized gene for the XYPylRS that recognizes BrEtY (Xiang *et al*, 2014) derived from *Methanosarcina mazei* PylRS (custom synthesized by GeneArt) under control of a CMV promoter, as well as four tandem repeats of the gene encoding for the enhanced pyrrolysine-tRNA$^{M15}$ (Serfling *et al*, 2018a). The tRNA gene is depleted of the CCA end and is driven by a U6 promoter and followed by a T-rich trailer.

### Cell culture

HEK293T cells were maintained in Dulbecco's modified Eagle's medium (DMEM; high glucose 4.5 g/l, 4 mM glutamine, pyruvate; Thermo Fisher Scientific) supplemented with 10% fetal calf serum (FCS; v/v; Thermo Fisher Scientific) and 100 U/ml penicillin and 100 µg/ml streptomycin (Thermo Fisher Scientific) (full DMEM) at 37°C under 5% CO$_2$ and 95% humidity. Cells were passaged at ~ 70–80% confluence.

### Photo-crosslinking experiments (Fig EV1)

HEK293T cells were seeded at 500,000 cells per well in six-well plates in 2 ml full DMEM. After 24 h, the media was exchanged with full DMEM supplemented with 250 µM *p*-benzoyl-phenylalanine (Bachem). Cells were transfected using PEI (Polysciences) at a ratio of PEI:DNA 3:1 (w/w) in lactate buffered saline (20 mM sodium lactate pH 4 and 150 mM NaCl) (Serfling & Coin, 2016; Serfling *et al*, 2018b). Cells were co-transfected with three plasmids: (i) 900 ng of a plasmid bearing the arrestin stop codon mutant, (ii) 900 ng of pIRE4-BpaRS, and (iii) 300 ng of a vector encoding the GPCR. Forty-eight hours post-transfection, the media was aspirated and exchanged with 1 ml activation buffer (PBS + 0.1% BSA). For the stimulation of the cells, the activation buffer was supplemented with 200 nM of the corresponding agonist (Ucn1 for CRF$_1$R, AVP for V$_2$R and PTH(1–34) for PTH1R). After 15 min at 37°C, the cells were irradiated with UV light for 15 min in a BLX-365 crosslinker (Bio-Budget Technologies, 365 nm; 4 × 8 W bulbs). Then, the activation buffer was aspirated and the cells were put at −80°C for 20–30 min, detached with 1 ml PBS supplemented with 1× protease inhibitor cocktail (Roche), and pelleted at 2,500 *g* for 10 min at 4°C. Cells were lysed in 80 µl Triton lysis buffer 1 (50 mM HEPES pH 7.5, 150 mM NaCl, 10% glycerol, 1% Triton X-100, 1.5 mM MgCl$_2$, 1 mM EGTA, 1 mM DTT and 1× protease inhibitor) for 30 min on ice and vortexed every 10 min. The samples were centrifuged at 16,000 *g* for 10 min at 4°C, and supernatants were transferred to pre-chilled tubes. For SDS–PAGE, 4 µl of supernatant was incubated for 30 min at 37°C in lithium dodecyl sulfate (LDS)-sample buffer

(250 mM Tris–HCl pH 8.5, 2% (w/v) LDS, 150 mM DTT, 0.4 mM EDTA, 10% (v/v) glycerol, and 0.2 mM Coomassie Brilliant Blue G). For deglycosylation, samples were treated with PNGaseF (New England Biolabs) according to the manufacturer's protocol and LDS-sample buffer was added before SDS–PAGE.

## Chemical crosslinking experiments

HEK293T cells were seeded at 500,000 cells per well in six-well plates in full DMEM. After 24 h, the media was exchanged with full DMEM supplemented with 250 μM BrEtY (synthesized as described in Xiang *et al* (2014)). Cells were transfected using PEI as described above. Cells were co-transfected with three plasmids: (i) 900 ng of a plasmid bearing the arrestin stop codon mutant, (ii) 900 ng of pNEU-MmXYRS-4xM15, and (iii) 300 ng of a vector encoding for PTH1R or a Ser-PTH1R mutant. Forty-eight hours later, the media was aspirated and the cells were stimulated for 90 min with 1 ml activation buffer supplemented with 200 nM PTH(1–34). Cell lysis and sample preparation for SDS–PAGE were carried out as described in the section about photo-crosslinking.

## SDS–PAGE and Western blot

Samples were resolved on 8% Tris/glycine polyacrylamide gels and transferred to a PDVF membrane (Millipore Immobilon, Merck, pore size 0.45 μm). The membranes were blocked by 5% non-fat dry milk powder (NFDM) in TBS-T (20 mM Tris–HCl pH 7.4, 150 mM NaCl and 0.1% (v/v) Tween-20) for at least 1 h at RT. Primary antibodies were diluted in 5 % NFDM in TBS-T as follows: α-HA-4D2 (Roche) 1:2,000; α-PTH1R 4D2 (Thermo Fisher Scientific) 1:2,000; α-FLAG-HRP M2 (Sigma-Aldrich) 1:5,000; α-1D4-HRP (Santa Cruz Biotechnology, sc-57432) 1:2,000. Membranes were incubated for either 1 h at RT (α-FLAG-HRP) or overnight at 4°C with the primary antibody (α-HA, α-PTH1R and α-1D4-HRP), followed by 3 × 10 min washes in TBS-T. Secondary antibodies, either α-rat-HRP (Cell Signaling) or α-mouse-HRP (Santa Cruz Biotechnology, sc-516102), were used at 1:10,000 in 5% NFDM in TBS-T for 1 h at RT followed by 3 × 10 min wash cycles in TBS-T. Membranes were soaked in homemade ECL reagent (10 parts 0.1 M Tris–HCl pH 8.6 with 250 mg/l luminol, one part DMSO with 1,100 mg/l *p*-hydroxycoumarin acid, and 0.003 parts 30% $H_2O_2$). After 1 min, signals were detected for 5 min in the dark (Gbox, Syngene).

## Immunoprecipitation

HEK293T cells were seeded in 15 cm dishes at $7 \times 10^6$ cells per dish. After 24 h, the media was exchanged with full DMEM supplemented with 250 μM Bpa. Cells were transfected using PEI as described above with 17.5 μg pcDNA3.1_Arr2-D78TAG-2xStrep and 17.5 μg of pIRE4-Bpa. After 48 h, the cells were irradiated with UV light, as described above. The media was aspirated, and the cells were frozen at −80°C for 20–30 min and detached with 5 ml Triton lysis buffer 2 (20 mM Tris–HCl pH 7.4, 150 mM NaCl, 1% (v/v) Triton X-100, and 1× protease inhibitor) per dish. After an incubation of 30 min on ice with vortexing steps every 10 min, the cell debris was pelleted at 16,000 *g* for 45 min at 4°C. The supernatant was supplemented with NaCl and urea for a final concentration of 1 M each. Strep-Tactin-XT beads (IBA Lifesciences) were added to the supernatant (1 μl of resin, i.e., 2 μl slurry per 1 ml of supernatant) and incubated overnight at 4°C under constant gentle agitation. The sample was loaded on a 1 ml empty column equipped with a filter and washed five times with one column volume (CV) of buffer W (100 mM Tris–HCl pH 8.0, 150 mM NaCl, and 1 mM EDTA). Bound protein was eluted with 8 × 1 CV Buffer BXT (Buffer W supplemented with 50 mM biotin). Samples of the supernatant, flow-through, wash, and elution fractions were separated by SDS–PAGE and analyzed by Western blot, as described above. Membranes were blocked overnight in 1% BSA in TBS-T and incubated overnight at 4°C in α-Strep-tag antibody (IBA Lifesciences; 1:2,000 in blocking solution). Secondary antibody incubation with α-mouse-HRP and ECL detection was performed as described above. All fractions containing the protein of interest were pooled and concentrated to a total volume of 500 μl using a centrifugal concentrator with a 3K cutoff membrane (PALL) according to the manufacturer instructions.

## Size exclusion chromatography

The purified crosslinked sample from the Strep-Tactin IP was applied to SEC. The SEC was performed at 4°C using ÄKTA advent system equipped with a Superdex 200 Increase 10/300 GL column (GE Healthcare). Buffer W was used as running buffer. For all steps the flowrate was set to 0.6 ml/min. The column was equilibrated with 1.5 CV of Buffer W. Absorbance was detected at 280 nm.

## MALDI-TOF-MS

After SEC, protein samples were embedded in "super-DHB" matrix, a 9:1 (w/w) mixture of 2,5-dihydroxybenzoic acid and 2-hydroxy-5-methoxybenzoic acid, and analyzed via MALDI-TOF-MS (Ultraflex III, Bruker Daltonik, Bremen).

## Nano-HPLC/Nano-ESI-Orbitrap-MS/MS

Samples from SEC were proteolyzed either by trypsin or a combination of trypsin and endoproteinase GluC with the SMART Digest trypsin kit (Thermo Fisher Scientific) following the manufacturer's protocol (1 h, 70°C). For the combined trypsin/GluC digestion, trypsin beads were removed after the tryptic digestion before GluC was added. Proteolysis was allowed to proceed for 2 h at 37°C. Then, cysteines were reduced with DTT and alkylated with iodoacetamide.

Samples were analyzed on an Ultimate 3000 RSLC nano-HPLC system coupled with an Orbitrap Q-Exactive Plus mass spectrometer (Thermo Fisher Scientific). Chromatography was performed by applying 90-min gradients with reversed phase C18 columns (μPAC 900 nl C18 trapping column and μPAC™ 50 cm C18 chip-based separation column, Pharmafluidics). For MS data acquisition, a data-dependent top 10 method was used. FTMS survey scans were performed in the *m/z* range 375–1,799, $R = 140,000$ at *m/z* 200, AGC (automated gain control) target value $3 \times 10^6$, and a maximum injection time of 100 ms. MS/MS scans were performed of the 10 most abundant signals of the survey scan with stepped higher energy collision-induced dissociation with 27, 30, and 33% normalized collision energy, quadrupole isolation window 2 Th, AGC target value $2 \times 10^5$, and maximum injection time 250 ms. Dynamic exclusion was enabled, and exclusion time was set to 60 s.

LC/MS data were processed with the Proteome Discoverer software, version 2.4 (Thermo Fisher Scientific). For peptide identification, MS/MS data were searched against the UniProt database (version Feb. 2019, taxonomy human, 73,801 entries, with sequence of arrestin-strep added) using Sequest HT with the following settings: Precursor mass error < 5 ppm, product ion mass error < 0.02 Da, variable modifications: Oxidation of Met, acetylation of protein N-termini, conversion of Asp to Bpa, fixed modification: carbamidomethylation of Cys, up to three missed cleavage sites. Peptides were filtered for false discovery rate < 1%.

**Recruitment assay**

Ligand-induced recruitment of βarr1/2 and their Bpa variants to PTH1R was assessed using the NanoBiT® complementation assay (Soave *et al*, 2020). The constructs for the assay were cloned as follows: PTH1R-LgBiT was cloned by inserting the ORF of the PTH1R into pBiT1.1-C using the restriction sites NheI and EcoRI. SmBiT-βarr1/2 were cloned by inserting the ORF of βarr1/2 in pBiT2.1-N between the restriction sites XhoI and NheI.

Ninety-six-well plates (Greiner) were precoated with poly-D-lysine (MW = 500–550 kDa) as described before (Serfling *et al*, 2018b). HEK293T cells were seeded at a density of 20,000 cells/well. After 24 h, half of the media was discarded and replaced with full DMEM supplemented with Bpa to a final concentration of 250 μM. The cells were transiently transfected with expression constructs for SmBit-βarr1/2-xxxTAG (50 ng), PTH1R-LgBit (50 ng), and pIRE4-Bpa (50 ng) using PEI as described above. After 48 h, the cells were washed with FluoroBrite media (Thermo Fisher Scientific) and subsequently kept in 100 μl FluoroBrite media (+10% FCS, 1× Pen/Strep). Twenty-five microliter per well of a 5× solution of the Nano-Glo® live cell reagent containing the cell-permeable furimazine substrate was added, and the baseline luminescence was immediately monitored for 15 min with 30-s intervals. The ligand PTH(1–34) was added to a final concentration of 200 nM, and luminescence was immediately monitored for 45 min with 30-s intervals. All luminescence measurements were performed using a FLUOstar Omega plate reader (BMG Labtech) at 37°C. Data analysis was performed using GraphPad Prism 5 (GraphPad Software).

## Data availability

The mass spectrometry proteomics data have been deposited to the ProteomeXchange Consortium via the PRIDE (Perez-Riverol *et al*, 2019) partner repository with the dataset identifier PXD020418 (http://www.ebi.ac.uk/pride/archive/projects/PXD020418) and https://doi.org/10.6019/PXD020418.

**Expanded View** for this article is available online.

## Acknowledgements
We thank John Heiker and Regina Reppich-Sacher for assistance with the MALDI-TOF-MS analysis in Leipzig, Karolin Dipper for help with the chemical synthesis of BrEtY, and Yasmin Aydin for the design of the graphical Synopsis and Fig EV1. This work was supported by the German Research Foundation (DFG Grants: Emmy-Noether Co822/2-1 and Co822/3-1 to IC, Si867/15-2 to AS), the Max-Buchner-Forschungsstiftung (Fellowship 3456 to IC), NIH grants R35 GM122491 and RO1 EY011500 to VVG, and Cornelius Vanderbilt Chair (VVG). Open access funding enabled and organized by Projekt DEAL.

## Author contributions
IC conceived and supervised the project. TB established the Bpa incorporation and crosslinking procedure, performed all crosslinking experiments, purified crosslinked products, performed preliminary MS analysis, performed part of the arrestin recruitment experiments, and synthesized BrEtY under the supervision of EB. SE performed the SEC purification and analysis; RS established the arrestin recruitment assays and performed part of recruitment experiments; CI performed MS analysis under the supervision of AS; and VVG participated to the conception of the project and provided material and advice. The manuscript was written by TB with the collaboration of SE and CI and was revised by IC and VVG.

## Conflict of interest
The authors declare that they have no conflict of interest.

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
