## [Review Process File · EMBO Reports]

Exploring GPCR-arrestin interfaces with genetically encoded crosslinkers

Thore Böttke, Stefan Ernicke, Robert Serfling, Christian Ihling, Edyta Burda, Vsevolod Gurevich, Andrea Sinz, and Irene Coin

DOI: [10.15252/embr.202050437](https://doi.org/10.15252/embr.202050437)

Corresponding author(s): Irene Coin (irene.coin@uni-leipzig.de), Thore Böttke (thore.boettke@uni-leipzig.de)

Review Timeline:

Submission Date:	16th Mar 20
Editorial Decision:	28th Apr 20
Revision Received:	10th Jul 20
Editorial Decision:	5th Aug 20
Revision Received:	10th Aug 20
Accepted:	13th Aug 20

Editor: Achim Breiling

Transaction Report:

Dear Prof. Coin,

Thank you for the submission of your research manuscript to EMBO reports. We have now received reports from the three referees that were asked to evaluate your study, which can be found at the end of this email.

As you will see, all referees think that the findings are of interest, but they also have several comments, concerns and suggestions, indicating that a major revision of the manuscript is necessary to allow publication in EMBO reports. As the reports are below, and I think all points need to be addressed, I will not detail them here.

Given the constructive referee comments, we would like to invite you to revise your manuscript with the understanding that all referee concerns must be addressed in the revised manuscript and/or in a detailed point-by-point response. Acceptance of your manuscript will depend on a positive outcome of a second round of review. It is EMBO reports policy to allow a single round of revision only and acceptance of the manuscript will therefore depend on the completeness of your responses included in the next, final version of the manuscript.

Revised manuscripts should be submitted within three months of a request for revision. We are aware that many laboratories cannot function at full efficiency during the current COVID-19/SARS-CoV-2 pandemic and we have therefore extended our 'scooping protection policy' to cover the period required for full revision. Please contact me to discuss the revision should you need additional time, and also if you see a paper with related content published elsewhere.

1) a .docx formatted version of the final manuscript text (including legends for main figures, EV figures and tables), but without the figures included. Please make sure that changes are highlighted to be clearly visible. Figure legends should be compiled at the end of the manuscript text.

2) individual production quality figure files as .eps, .tif, .jpg (one file per figure), of main figures and EV figures. Please upload these as separate, individual files upon re-submission.

The Expanded View format, which will be displayed in the main HTML of the paper in a collapsible format, has replaced the Supplementary information. You can submit up to 5 images as Expanded View. Please follow the nomenclature Figure EV1, Figure EV2 etc. The figure legend for these should be included in the main manuscript document file in a section called Expanded View Figure Legends after the main Figure Legends section. Additional Supplementary material should be supplied as a single pdf file labeled Appendix. The Appendix should have page numbers and needs

to include a table of content on the first page (with page numbers) and legends for all content. Please follow the nomenclature Appendix Figure Sx, Appendix Table Sx etc. throughout the text, and also label the figures and tables according to this nomenclature.

For more details please refer to our guide to authors:

See also our guide for figure preparation:

http://wol-prod-cdn.literatumonline.com/pb-assets/embosite/EMBOPress_Figure_Guidelines_061115-1561436025777.pdf

4) a complete author checklist, which you can download from our author guidelines (<https://www.embopress.org/page/journal/14693178/authorguide>). Please insert page numbers in the checklist to indicate where the requested information can be found in the manuscript. The completed author checklist will also be part of the RPF.

Please also follow our guidelines for the use of living organisms, and the respective reporting guidelines: <http://www.embopress.org/page/journal/14693178/authorguide#livingorganisms>

5) that primary datasets produced in this study (e.g. RNA-seq, ChIP-seq and array data) are deposited in an appropriate public database. This is now mandatory (like the COI statement). If no primary datasets have been deposited in any database, please state this in this section (e.g. 'No primary datasets have been generated and deposited').

The accession numbers and database should be listed in a formal "Data Availability " section (placed after Materials & Methods) that follows the model below. Please note that the Data Availability Section is restricted to new primary data that are part of this study.

Data availability

6) We strongly encourage the publication of original source data with the aim of making primary data more accessible and transparent to the reader. The source data will be published in a separate source data file online along with the accepted manuscript and will be linked to the relevant figure. If you would like to use this opportunity, please submit the source data (for example scans of entire gels or blots, data points of graphs in an excel sheet, additional images, etc.) of your key experiments together with the revised manuscript. If you want to provide source data, please include size markers for scans of entire gels, label the scans with figure and panel number, and send one PDF file per figure.

8) Regarding data quantification and statistics, can you please specify, where applicable, the number "n" for how many independent experiments (biological replicates) were performed, the bars and error bars (e.g. SEM, SD) and the test used to calculate p-values in the respective figure legends. Please provide statistical testing where applicable, and also add a paragraph detailing this to the methods section. See: <http://www.embopress.org/page/journal/14693178/authorguide#statisticalanalysis>

9) Please provide 5 keywords on the title page.

I look forward to seeing a revised version of your manuscript when it is ready. Please let me know if you have questions or comments regarding the revision.

Yours sincerely,

Achim Breiling
Editor
EMBO Reports

Referee #1:

Boettke et al. report a study aimed at elucidating the dynamics of GPCR-beta arrestin interactions in live cells using a biochemical/photochemical cross linking approach. They incorporate photoactive non-natural amino acids into expressed beta-arrestin 1 or beta-arrestin 2 and then look at cross linking (for example +/- UV light, +/- agonist ligand) for three different GPCRs (vasopressin 2R, PTH1R, CRF1R). They focus mainly on the so-called finger loop region of the arrestin. Each different potential crosslink represents a different site directed arrestin mutation (to amber codon) which allows some "mapping" of the crosslinking sites when cross linking occurs versus sites that do not show crosslinking. The results are more a proof-of-concept than an advance in understanding to GPCR-arrestin interaction mechanism, but the approach is potentially useful if additional data from other receptor-arrestin pairs can be accumulated.

The paper is short and the results and discussion need to be enhanced. For example, the three GPCRs chosen for study are not mentioned in the abstract or introduction. It is not explained why or how the three receptors were chosen. There is no mention of the fact that GPCRs are classified into two classes with respect to arrestin coupling (Luttrell and Lefkowitz, J Cell Sci 115:455, 2002 and multiple papers from M. Bouvier lab). Class A show weak transient interactions (like B2 adrenergic receptor) while class B show strong long-lasting interactions (like angiotensin II type 1 receptor) as described earlier. The best use of the current would be to compare receptors from each class side by side. It is not clear from Supple Fig 1 whether PTH1R is a class A or class B. In any case, this whole concept is absent from the paper. In addition, a recent relevant paper (Gagnon et al J Biol Chem 2019) is cited only in passing, and probably should be described in more detail since they used a similar method, only with crosslinkers on the GPCR and not the arrestins.

In summary, an interesting paper with enough data to support the application of the method more widely. However, the current paper could be much improved by a more detailed discussion of the relevant literature and questions that could be addressed -most obviously the differences between class A and class B GPCR arrestin interactions.

Referee #2:

The present manuscript "Exploring GPCR-arrestin interfaces with genetically encoded crosslinkers" by I. Coin and colleagues try to characterize the complex binding patterns (here especially the finger loop region) of arrestins (here, b-arr1 (arr-2) and b-arr2 (arr-3)) to G-protein coupled receptor (GPCR) in more detail. Many biochemical studies and also the available structural data suggest that the complexation and also the binding process could indeed function somewhat differently depending on the receptor. This might be due to the broad versatility of the arrestin, but also to the specific imprint or topology of the receptors. The presented study tries to answer this question with a newly established very interesting method of incorporation of non-canonical amino acids (ncAAs) for photo- and chemical cross-linking.

The very short paper reads very technical, but the methods seem to be very sound. I think it is very important to find new ways to get the complex information about the pharmacologically important interactions between arrestin and GPCRs. Unfortunately, we see different states in the 3D structures that we have, which can be due to the different preparations of the complexes or different environments for the complexes (e.g. nanodiscs, lipids, crystal contacts etc.). All external influences are there a kind of bias for the complex formation. But maybe it is also due to the manifold charged surface, rotatability and oligomizability of the arrestins and the intracellular diversity of the receptor loops.

In summary, in addition to structures or in vitro spectroscopic methods, we also need in vivo approaches where we can analyze an almost native complex formation. This approach contributes to this topic. It will now be important to follow this approach further and to photo or chemical-crosslink even more places in the whole arrestin in more detail. In addition, it should now also be possible to test it on different receptors, including those that only bind one arrestin very specifically.

Remarks:

- The authors should definitely create a general figure in which they show schematically the procedure of the ncAA assay. It would be better to illustrate it for the reader in a figure.

- Can reaction with other GPCRs in HEK be definitely excluded, which occur naturally?
- What does the band (smear band?) under the prominent band (b-arrestin - or + Y63ncAA) in Figure 1C mean? Degradation?
- Figure S1 is very difficult to read and distinguish.
- It sounds plausible and was nicely shown (Fig.3) that these could be arrestin dimers (in Figure 2D, D78 etc.), which are visible between 100-200kDa. However, the prerequisite is that there are no native receptors (in HEK) that could give a signal. Or could they be derived from other receptors? However, in some spots in Figures 1A-C there are bands without the addition of agonists. What is the reason for this? In some examples, there are significantly more than in Fig. 2D (e.g. Figure 2B, L71/G72/Y63 or Figure 2A D68/L69 etc).
- I recommend to make a supplementary figure to illustrate this arrestin dimer interface.

 Referee #3:

In this manuscript Coin et al. used genetically encoded photo- and chemical crosslinkers to probe the interaction of two b-arrestins with several GPCRs in live mammalian cells. b-arrestins regulate GPCR signaling. Although recently the structures of several b-arrestin/GPCR complexes have been solved, many others remain elusive due to the technical challenge. The approach described in this work enables probing GPCR/b-arrestin in live mammalian cells, complementing the in vitro structural studies and affording results that can be more physiologically relevant. More specifically, the authors found that 1) each GPCR crosslinked with b-arrestin with distinct patterns; 2) the two b-arrestins showed similar patterns; 3) b-arrestin existed as a dimer; and 4) using chemical crosslinker BetY, inter-molecular proximity points of arrestin-receptor were identified, which is compatible with the orientation of b-arrestin relative to the receptor in the barr1-Rho structure but not in the barr1-NTS1R structure. This type of in cell crosslinking work is very challenging, and it is thus exciting to see these results. Overall, this manuscript provide interesting and significant results on the b-arrestin/GPCR interaction obtained in the context of mammalian cells. The experiments are well designed and support the claims. Currently there are relatively few reports on probing receptor-ligand interaction directly in live cells, which makes this work stand out. These results will be broadly interesting to researchers working on GPCRs, signaling, and drug design. I therefore support publication after the following minor points are addressed:

- 1) page 3: "The Bpa-mutation did not hamper the receptor-arrestin interaction: all arrestins were recruited within a few minutes to the parathyroid hormone receptor (PTH1R), which is known to recruit both b-arrestins". The recruiting experiment was done with PTH1R, so concluding "the Bpa-mutation did not hamper the receptor-arrestin interaction" in a general term for all receptors may be too strong, unless further explanation can justify it.
- 2) Figure S1: error bars for technical replicates only indicate pipetting errors or assay errors, which are not very useful for determining difference from WT. It will be more useful to plot the errors of independent experiments.
- 3) A highlight of this work is the chemical crosslinking of interacting proteins in live cells using

genetically encoded chemical crosslinkers. One such earlier work may worth citing when introducing BrEtY: Nat. Commun. 2017; 8(1):2240. PMID: 29269770.

4) b-arrestin presumably will interact with other proteins in cells in addition to the GPCR. The authors also observed some bands in the controls shown in Figure 2D, with some bands already nicely figured out (i.e., b-arrestin intra-molecular crosslinking and dimerization). It may be interesting to briefly mention other possibilities in discussion. Note I do not ask for experimental proof here, given the main focus of b-arrestin/GPCR interaction.

5) Page 5, line 2: Supplementary Figure 5 should be 4.

Referee #1:

Boettke et al. report a study aimed at elucidating the dynamics of GPCR-beta arrestin interactions in live cells using a biochemical/photochemical cross linking approach. They incorporate photoactive non-natural amino acids into expressed beta-arrestin 1 or beta-arrestin 2 and then look at cross linking (for example +/- UV light, +/- agonist ligand) for three different GPCRs (vasopressin 2R, PTH1R, CRF1R). They focus mainly on the so-called finger loop region of the arrestin. Each different potential crosslink represents a different site directed arrestin mutation (to amber codon) which allows some "mapping" of the crosslinking sites when cross linking occurs versus sites that do not show crosslinking. The results are more a proof-of-concept than an advance in understanding to GPCR-arrestin interaction mechanism, but the approach is potentially useful if additional data from other receptor-arrestin pairs can be accumulated.

In summary, an interesting paper with enough data to support the application of the method more widely. **However, the current paper could be much improved by a more detailed discussion of the relevant literature and questions that could be addressed -most obviously the differences between class A and class B GPCR arrestin interactions.**

As described below, we have expanded the discussion of relevant literature and made of the class A/B concept a strong point in the paper.

Specific Comments:

- 1. The paper is short and the results and discussion need to be enhanced. For example, the three GPCRs chosen for study are not mentioned in the abstract or introduction. It is not explained why or how the three receptors were chosen.**

Thank you! In revised manuscript we have mentioned the different GPCRs tested in the abstract. In the introduction, we clearly described their behavior toward arrestin. In the results section, we explain why we have chosen these three receptors. All these points are explained in the frame of the class A/B concept, please see details in the following point.

- 2. There is no mention of the fact that GPCRs are classified into two classes with respect to arrestin coupling (Luttrell and Lefkowitz, J Cell Sci 115:455, 2002 and multiple papers from M. Bouvier lab). Class A show weak transient interactions (like B2 adrenergic receptor) while class B show strong long-lasting interactions (like angiotensin II type 1 receptor) as described earlier [...]. In any case, this whole concept is absent from the paper.**

We are truly grateful to the reviewer for this comment, which made us realize the importance of the concept of class A and B interactions in our work. We have now added a whole paragraph about class A and class B arrestin binders in the introduction, and cited the relevant literature on this concept. In this paragraph, we also mention the three receptors that we have investigated in the study and describe their behavior toward arrestin:

GPCRs are divided into two classes with respect to arrestin binding (Luttrell & Lefkowitz, 2002; Oakley *et al*, 1999; Oakley *et al*, 2000). Class A receptors form transient and rapidly dissociating complexes with arrestin, and resensitize rapidly. These receptors interact with both β -arrestins, but show a bias toward β arr2. Besides the prototypical β 2-adrenergic, class A receptors include among others the μ opioid, endothelin A and dopamine D1A receptors (all rhodopsin-like GPCRs), as well as the corticotropin releasing factor receptor (CRF₁R, secretin-like) (Grammatopoulos, 2012; Oakley *et al*, 2007). Class B receptors engage arrestin in long-lived complexes that remain associated during internalization via clathrin-coated pits, and resensitize slowly. Class B receptors bind with high affinity either β -arrestin. The prototypic class B arrestin binder is the vasopressin 2 receptor (V₂R). Other receptors forming stable complexes with arrestin include the angiotensin II

type 1 receptor (AT1R), the oxytocin receptor, the neurotensin 1 receptor (NTS₁R) and the secretin-like parathyroid hormone receptor (PTH1R) (Oakley *et al*, 2001; Vilardaga *et al*, 2002). The latter has been shown to form GPCR-arrestin-G-protein megaplexes that mediate prolonged signaling after internalization in endosomes (Thomsen *et al*, 2016; Wehbi *et al*, 2013). A major determinant for the stability of arrestin-GPCR complexes is the C-terminal tail of the receptor. In general, GPCRs carrying clusters of Ser/Thr in the C-terminal tail show a class B behavior (Oakley *et al*, 1999).

We have also mentioned the arrestin binding behavior of the GPCRs for which structures of the receptor-arrestin complex has been solved:

The β_1 -AR belongs to class A arrestin binders (Eichel, Jullié *et al*, 2016, Shiina, Kawasaki *et al*, 2000), whereas contradictory findings have been reported for the M₂R (Gurevich, Dion *et al*, 1995, Jones, Echeverry *et al*, 2006).

We then recall the concept of A and B class at the beginning of the results section. In this way, we also justify the choice of our receptors:

The two sets of Bpa- β arr1 and Bpa- β arr2 were combined with GPCRs forming either stable (class B) or transient (class A) complexes with arrestins. We selected two class B receptors belonging to two distinct phylogenetic GPCR families, the prototypical V₂R (rhodopsin-like) and the PTH1R (secretin-like), as well as the class A receptor CRF₁R (secretin-like). The latter was preferred to other class A GPCRs because the CRF system is well established in our laboratory.

Last, we discuss our findings with respect to the class A/B concept in the conclusion:

We show that different footprints are not related to the phylogenetic class of the investigated GPCRs (rhodopsin-like vs. secretin like) nor to the type of interaction with arrestin (transient or stable), but are indeed specific for each receptor. [...] We have shown that the approach is applicable both to stable and transient arrestin complexes, which provides a unique possibility for investigating interactions of arrestin with class A GPCRs without the need of generating chimeras with altered binding behavior.

3. The best use of the current would be to compare receptors from each class side by side. It is not clear from Supple Fig 1 whether PTH1R is a class A or class B.

We thank the reviewer for this very helpful comment. In fact, we used receptors of both classes in our work. We have now emphasized this comparison in the text (as explained in point 2):

The two sets of Bpa- β arr1 and Bpa- β arr2 were combined with GPCRs forming either stable (class B) or transient (class A) complexes with arrestins.

We have clearly explained that PTH1R is a class B and also described the formation of arrestin megaplexes that has been reported for this receptor (as mentioned in point 2):

Other receptors forming stable complexes with arrestin include [...]and the secretin-like parathyroid hormone receptor (PTH1R) (Oakley *et al*, 2001; Vilardaga *et al*, 2002). The latter has been shown to form GPCR-arrestin-G-protein megaplexes that mediate prolonged signaling after internalization in endosomes (Thomsen *et al*, 2016; Wehbi *et al*, 2013).

We have also highlighted an important result of our work respect to transient and stable arrestin complexes: This result clearly confirms the observation from the existing structures that the orientation of arrestin on the receptor does not determine whether the GPCR-arrestin complexes are stable or transient.

4. In addition, a recent relevant paper (Gagnon et al J Biol Chem 2019) is cited only in passing, and probably should be described in more detail since they used a similar method, only with crosslinkers on the GPCR and not the arrestins.

We agree with the reviewer that this paper should be described more in detail. We have added following paragraph in the discussion.

In a recent report, photo-crosslinkers genetically incorporated into the intracellular domains of the AT₁R have revealed distinct binding modalities of AT₁R to β arr1 depending on the type of the agonist used for its activation (natural angiotensin vs. arrestin-biased AT₁R ligands) (Gagnon *et al*, 2019). Altogether, these results demonstrate that genetically encoded photo-crosslinkers incorporated either into a GPCR or into arrestin allow elucidating with a good sensitivity differences in the arrangement of arrestin-GPCR complexes.

Referee #2:

The present manuscript "Exploring GPCR-arrestin interfaces with genetically encoded crosslinkers" by I. Coin and colleagues try to characterize the complex binding patterns (here especially the finger loop region) of arrestins (here, b-arr1 (arr-2) and b-arr2 (arr-3)) to G-protein coupled receptor (GPCR) in more detail. Many biochemical studies and also the available structural data suggest that the complexation and also the binding process could indeed function somewhat differently depending on the receptor. This might be due to the broad versatility of the arrestin, but also to the specific imprint or topology of the receptors. The presented study tries to answer this question with a newly established very interesting method of incorporation of non-canonical amino acids (ncAAs) for photo- and chemical cross-linking.

The very short paper reads very technical, but the methods seem to be very sound. I think it is very important to find new ways to get the complex information about the pharmacologically important interactions between arrestin and GPCRs. Unfortunately, we see different states in the 3D structures that we have, which can be due to the different preparations of the complexes or different environments for the complexes (e.g. nanodiscs, lipids, crystal contacts etc.). All external influences are there a kind of bias for the complex formation. But maybe it is also due to the manifold charged surface, rotatability and oligomizability of the arrestins and the intracellular diversity of the receptor loops.

In summary, in addition to structures or in vitro spectroscopic methods, we also need in vivo approaches where we can analyze an almost native complex formation. This approach contributes to this topic. It will now be important to follow this approach further and to photo or chemical-crosslink even more places in the whole arrestin in more detail. In addition, it should now also be possible to test it on different receptors, including those that only bind one arrestin very specifically.

Remarks:

1. The authors should definitely create a general figure in which they show schematically the procedure of the ncAA assay. It would be better to illustrate it for the reader in a figure.

Thank you for this suggestion! We have included a schematic overview of the photo crosslinking assay as Expanded View Figure 1.

Figure EV1: β arr-GPCR photo-crosslinking. A) 1: Cultured HEK293T cells are supplemented with 250 μ M Bpa and transfected with three plasmids. The first plasmid carries the ORF for a GPCR, the second plasmid carries the ORF for one β arr TAG-mutant and the third plasmid encodes for the tRNA/synthetase pair that incorporates Bpa. 2: The GPCR and the β arr-xxxBpa variant are expressed for 48h. 3: The GPCR is activated with an agonist for 15 min. 4: The cells are irradiated with UV-light (365 nm) for 15 min 5: Samples are lysed, resolved on SDS-PAGE, and the crosslinked arrestin-GPCR complex is detected by immunostaining. B) Mechanism of photo-activation of Bpa. If the diradical species is in close proximity to the GPCR, crosslinking can occur. In general, Bpa inserts into C-H bonds (Dorman & Prestwich, 1994). A covalent arrestin-GPCR complex is formed.

2. Can reaction with other GPCRs in HEK be definitely excluded, which occur naturally?

This is an important point. We have different arguments that exclude the interference of endogenous receptors in our experiments. *First*, the signals of the receptor-arrestin complex in Western blot disappear when our overexpressed receptor is not activated (Figure 2), showing that arrestin is not recruited without activation. The three receptors investigated here are peptide-receptors that are activated by ligands with very high specificity, and our results further indicate that no other receptors are activated in our experiments. *Second*, crosslinking bands are specific in size and position for each overexpressed receptor. *Third*, we have an experiment (Figure 2D) in which the Bpa-arrestins are activated in the absence of an overexpressed receptor. In this case, we see only very few crosslinking bands, which maintain the same MW after deglycosylation, suggesting that they do not include a GPCR or another membrane protein.

3. What does the band (smear band?) under the prominent band (b-arr1 - or + Y63ncAA) in Figure 1C mean? Degradation?

We agree that the halo under the major β arr1 bands is sub-optimal. As we use protease inhibitors in all samples and do all work on ice, it is quite unlikely that this “smear” is due to degradation of the protein. Indeed, to address this issue, we have run different SDS-PAGE gels with the same samples but loading different amounts of protein. We have observed that larger protein loads lead to a halo of the (strong!) β arr1 signal. This appears like the smear that is visible in Figure 1. Such halos disappear when adjusting the exposure time of the blot. However, we have chosen to include this image in the paper because by lower gel loading or by shorter exposure we could not see simultaneously the bands of the overexpressed and the endogenous arrestins. To detect all arrestins in the system, we used an α - β arrestin1/2 antibody. The endogenous arrestin runs right below the overexpressed β arr1, which is slightly larger due to its 3xHA affinity tag.

4. Figure S1 is very difficult to read and distinguish.

We thank the reviewer for this comment. Yes, the supplementary Figure 1 (now appendix Figure S1) reporting so many curves in the same graph was not clear. We have now partitioned the mutants in 12 different graphs, each reporting only two curves compared to wild type. We believe that the figure is much clearer now.

5. It sounds plausible and was nicely shown (Fig.3) that these could be arrestin dimers (in Figure 2D, D78 etc.), which are visible between 100-200kDa. However, the prerequisite is that there are no native receptors (in HEK) that could give a signal. Or could they be derived from other receptors?

As explained in point 2 above, we are very confident that there are no native receptors giving crosslinking signals in our experiments. Indeed, crosslinking bands that are activation independent do not respond to deglycosylation, which likely excludes the presence of a receptor. The MS analysis found no evidence for the presence of any GPCR in the D78Bpa dimer sample. Actually, the sample did not contain any other protein than β arr1, which had noteworthy peptide spectral matches. As we wrote in the results and discussion the score for β arr1 was 1,951 PSMs (peptide spectral matches) compared to only 22 PSMs for the second most abundant protein.

6. However, in some spots in Figures 1A-C there are bands without the addition of agonists. What is the reason for this? In some examples, there are significantly more than in Fig. 2D (e.g. Figure 2B, L71/G72/Y63 or Figure 2A D68/L69 etc).

Thank you for the comment. Yes, there are some bands in the non-activated activated samples, which are not always reproducible in the same intensity and could hint at a crosslinking with other endogenous proteins. It is indeed well known that β -arrestins interact with a huge variety of proteins. We have experimentally proved that all these receptor-independent bands do not respond to deglycosylation and therefore are unlikely to involve membrane proteins. We intend to further analyze all these bands using immunoprecipitation and MS analysis, both in live cells and using isolated arrestins in vitro. As this further work will require a lot of experimental effort and time, we believe that this accurate characterization is beyond the scope of this manuscript. We have added about these bands the following paragraph in the text:

Oh the other hand, it cannot be excluded that at least some of the weak activation-independent crosslinking signals at high MW belong to complexes of arrestin with endogenous proteins. Both arrestins are known to function as scaffolds for a wide variety of proteins, with the most prominent examples being kinases like ERKs, JNK3 or other MAPKs (Song *et al*, 2009; Xiao *et al*, 2007), as well as proteins involved in GPCR trafficking, such as clathrin and AP2 (Goodman *et al*, 1996; Laporte *et al*, 1999). Clearly, elucidating the nature of all receptor-independent crosslinking signals awaits further experiments.

7. I recommend to make a supplementary figure to illustrate this arrestin dimer interface.

Thank you for the suggestion, we agree that such a figure would be helpful. Sadly, we do not have enough data about the dimer interaction surface to make a meaningful model for the arrestin dimer. We only have one crosslinking hit where the crosslinking site for the second β arr1 is not known. We do plan to further investigate the dimerization interface, but this is another full project.

Referee 3:

In this manuscript Coin et al. used genetically encoded photo- and chemical crosslinkers to probe the interaction of two b-arrestins with several GPCRs in live mammalian cells. b-arrestins regulate GPCR signaling. Although recently the structures of several b-arrestin/GPCR complexes have been solved, many others remain elusive due to the technical challenge. The approach described in this work enables probing GPCR/b-arrestin in live mammalian cells, complementing the in vitro structural studies and affording results that can be more physiologically relevant. More specifically, the authors found that 1) each GPCR crosslinked with b-arrestin with distinct patterns; 2) the two b-arrestins showed similar patterns; 3) b-arrestin existed as a dimer; and 4) using chemical crosslinker BrEtY, inter-molecular proximity points of arrestin-receptor were identified, which is compatible with the orientation of b-arrestin relative to the receptor in the barr1-Rho structure but not in the barr1-NTS1R structure. This type of in cell crosslinking work is very challenging, and it is thus exciting to see these results. Overall, this manuscript provide interesting and significant results on the b-arrestin/GPCR interaction obtained in the context of mammalian cells. The experiments are well designed and support the claims. Currently there are relatively few reports on probing receptor-ligand interaction directly in live cells, which makes this work stand out. These results will be broadly interesting to researchers working on GPCRs, signaling, and drug design. I therefore support publication after the following minor points are addressed:

1) page 3: "The Bpa-mutation did not hamper the receptor-arrestin interaction: all arrestins were recruited within a few minutes to the parathyroid hormone receptor (PTH1R), which is known to recruit both b-arrestins". The recruiting experiment was done with PTH1R, so concluding "the Bpa-mutation did not hamper the receptor-arrestin interaction" in a general term for all receptors may be too strong, unless further explanation can justify it.

We thank the reviewer for this comment! We fully agree. We have softened the statement about the effect of the Bpa mutation and focused on the PTHR. The sentence was changed to: "The Bpa-mutation did not hamper the recruitment of either arrestin to the PTH1R receptor, suggesting that the overall functionality of the arrestins is preserved"

2) Figure S1: error bars for technical replicates only indicate pipetting errors or assay errors, which are not very useful for determining difference from WT. It will be more useful to plot the errors of independent experiments.

Thank you for the comment. Of course the reviewer is right that error bars for technical replicates are not really useful here. We included the error bars of the biological triplicates and clearly explained in the legend: Plotted data represent the arithmetic average of three independent experiments, each run in triplicate. Error bars represent the S.E.M. of the biological triplicates.

3) A highlight of this work is the chemical crosslinking of interacting proteins in live cells using genetically encoded chemical crosslinkers. One such earlier work may worth citing when introducing BrEtY: Nat. Commun. 2017; 8(1):2240. PMID: 29269770.

We thank the reviewer for reminding us of this important publication that we had not cited. We have now explicitly mentioned this paper in the results section: We further explored whether chemical crosslinking can be applied to determine inter-molecular arrestin-receptor proximity points as it was shown to capture protein-protein interactions in live cells (Yang *et al*, 2017).

4) b-arrestin presumably will interact with other proteins in cells in addition to the GPCR. The authors also observed some bands in the controls shown in Figure 2D, with some bands already nicely figured out (i.e., b-arrestin intra-molecular crosslinking and dimerization). It may be

interesting to briefly mention other possibilities in discussion. Note I do not ask for experimental proof here, given the main focus of b-arrestin/GPCR interaction.

We thank the reviewer for the comment. We have now discussed this possibility in the results and discussion section. Oh the other hand, it cannot be excluded that at least some of the weak activation-independent crosslinking signals at high MW belong to complexes of arrestin with endogenous proteins. Both arrestins are known to function as scaffolds for a wide variety of proteins, with the most prominent examples being kinases like ERKs, JNK3 or other MAPKs (Song *et al*, 2009; Xiao *et al*, 2007), as well as proteins involved in GPCR trafficking, such as clathrin and AP2 (Goodman *et al*, 1996; Laporte *et al*, 1999). Clearly, elucidating the nature of all receptor-independent crosslinking signals awaits further experiments.

5) Page 5, line 2: Supplementary Figure 5 should be 4.

We thank the reviewer for spotting this mistake. Supplementary figures have now been reorganized according to the EMBO guidelines.

Dear Prof. Coin

Thank you for the submission of your revised manuscript to our editorial offices. We have now received the reports from the three referees that were asked to re-evaluate your study, you will find below. As you will see, the referees now support the publication of your study in EMBO reports.

Before we can proceed with formal acceptance, I have these final editorial requests:

- It seems presently there is no specific call out for Figures 4A and 4B. Please add these to the manuscript text.
- Please remove the referee access information from the data availability section (DAS), and make sure the deposited data get public and are accessible using the information provided in the DAS upon publication of the paper.
- Finally, please find attached a word file of the manuscript text (provided by our publisher) with changes we ask you to include in your final manuscript text, and some queries, we ask you to address. Please provide your final manuscript file with track changes, in order that we can see any modifications done.

In addition I would need from you:

- a short, two-sentence summary of the manuscript
- two to three bullet points highlighting the key findings of your study

Kind regards,

Achim Breiling
Editor
EMBO Reports

Referee #1:

The manuscript has been dramatically improved. The work is now described much more clearly and the relevance of the findings to understanding GPCR-beta-arrestin interactions is significant. I recommend publication without delay.

Referee #2:

The revised new version of the manuscript is significantly refined. The authors fulfill all my suggestions for improvement. I recommend the acceptance of the manuscript.

Referee #3:

In the revised manuscript the authors have address my questions and comments satisfactorily. I support the publication of this work!

Authors made the requested editorial changes

Prof. Irene Coin
University of Leipzig
Institute of Biochemistry
Bruederstr. 34
Leipzig 04103
Germany

Dear Prof. Coin,

I am very pleased to accept your manuscript for publication in the next available issue of EMBO reports. Thank you for your contribution to our journal.

At the end of this email I include important information about how to proceed. Please ensure that you take the time to read the information and complete and return the necessary forms to allow us to publish your manuscript as quickly as possible.

As part of the EMBO publication's Transparent Editorial Process, EMBO reports publishes online a Review Process File to accompany accepted manuscripts. As you are aware, this File will be published in conjunction with your paper and will include the referee reports, your point-by-point response and all pertinent correspondence relating to the manuscript.

If you do NOT want this File to be published, please inform the editorial office within 2 days, if you have not done so already, otherwise the File will be published by default [contact: emboreports@embo.org]. If you do opt out, the Review Process File link will point to the following statement: "No Review Process File is available with this article, as the authors have chosen not to make the review process public in this case."

Should you be planning a Press Release on your article, please get in contact with emboreports@wiley.com as early as possible, in order to coordinate publication and release dates.

Thank you again for your contribution to EMBO reports and congratulations on a successful publication. Please consider us again in the future for your most exciting work.

Yours sincerely,

Achim Breiling
Editor
EMBO Reports

THINGS TO DO NOW:

You will receive proofs by e-mail approximately 2-3 weeks after all relevant files have been sent to

our Production Office; you should return your corrections within 2 days of receiving the proofs.

Please inform us if there is likely to be any difficulty in reaching you at the above address at that time. Failure to meet our deadlines may result in a delay of publication, or publication without your corrections.

All further communications concerning your paper should quote reference number EMBOR-2020-50437V3 and be addressed to emboreports@wiley.com.

Should you be planning a Press Release on your article, please get in contact with emboreports@wiley.com as early as possible, in order to coordinate publication and release dates.

Corresponding Author Name: Irene Coin

Manuscript Number: 50437-T